# Influenza Vaccination and Non-Pharmaceutical Measure Effectiveness for Preventing Influenza Outbreaks in Schools: A Surveillance-Based Evaluation in Beijing

**DOI:** 10.3390/vaccines8040714

**Published:** 2020-12-01

**Authors:** Ying Sun, Peng Yang, Quanyi Wang, Li Zhang, Wei Duan, Yang Pan, Shuangsheng Wu, Huaqing Wang

**Affiliations:** 1Department of National Immunization Program, Chinese Center for Disease Control and Prevention, Beijing 100050, China; sun1ying2@163.com; 2Institute for Infectious Diseases and Endemic Diseases Prevention and Control, Beijing Center for Disease Prevention and Control (CDC), Beijing 100013, China; bjcdcxm@126.com (Q.W.); zhangli-1219@163.com (L.Z.); duanwei_bjcdc@126.com (W.D.); panyang@bjcdc.org (Y.P.); wushuangsheng0606@126.com (S.W.); 3Office of Beijing Center for Global Health, Beijing Center for Diseases Prevention and Control (CDC), Beijing 100013, China; yangp@bjcdc.org; 4School of Public Health, Capital Medical University, Beijing 100069, China

**Keywords:** influenza vaccination, febrile outbreak, non-pharmaceutical measure, multi-level model, multi-stage regression

## Abstract

Although schools are known to play a major role in the spread of influenza virus, few studies have evaluated the effectiveness of vaccination and non-pharmaceutical measures for preventing influenza outbreaks in schools. We investigated all febrile illness outbreaks in primary and secondary schools in Beijing reported between August 2018 and July 2019. We obtained epidemiological information on febrile illness outbreaks and oral pharyngeal swabs from students in the outbreaks to test for influenza virus. We surveyed schools that did not report febrile illness outbreaks. We developed multi-level models to identify and evaluate factors associated with serious influenza outbreaks and explored the association of vaccine coverage and outbreaks using multi-stage regression models. We identified a total of 748 febrile illness outbreaks involving 8176 students in Beijing; 462 outbreaks were caused by influenza virus. Adjusted regression modeling showed that large class size (odds ratio (OR) = 2.38) and the number of days from identification of the first case to initiation of an intervention (OR = 1.17) were statistically significant and positively associated with serious outbreaks, and that high vaccination coverage (relative risk (RR) = 0.50) was statistically significant and negatively associated with outbreaks. Multi-stage regression modeling showed that RR decreased fastest when vaccination coverage was 45% to 51%. We conclude that high influenza vaccination coverage can prevent influenza outbreaks in schools and that rapid identification of febrile children and early initiation of non-pharmaceutical measures can reduce outbreak size.

## 1. Introduction

Seasonal influenza is one of the most common acute respiratory infectious diseases, annually affecting an estimated 5% to 10% of adults and 20% to 30% of children, causing over 290 thousand deaths worldwide every year, and resulting in a significant global health and economic burden [1,2]. Because of its high transmission ability, influenza virus can cause febrile outbreaks in crowded areas [3]. Schools provide confined spaces and frequent close contact that facilitate transmission. Combined with prolonged viral shedding and less historically-acquired immunity among school-age children that can accelerate the spread of influenza, outbreaks in schools can rapidly become severe [4,5,6,7]. Children are thought to be the main introducers of influenza into households, playing a major role in spreading influenza to households and into communities [4,8,9]. Primary and secondary schools are frequently affected by influenza febrile outbreaks in China [10,11]. Preventing and reducing school-based influenza outbreaks should, therefore, directly benefit children and indirectly benefit communities.

Influenza vaccination is recommended by the World Health Organization (WHO) and several countries/regions to prevent influenza infection and reduce its severe outcomes [12,13,14]. Since 2007, a school-based influenza vaccination program has been implemented in Beijing, in which inactive influenza vaccine (IIV) has been provided to primary and secondary school students at no charge to families. The intent of the program is to prevent influenza outbreaks and spread in communities. A febrile illness outbreak surveillance system have been established since 2006 to monitor, characterize, and respond to febrile illness outbreaks. Non-pharmaceutical interventions are used in an attempt to interrupt virus transmission in reported school outbreaks.

Previous studies have evaluated the effectiveness of influenza vaccines and non-pharmaceutical measures for preventing and mitigating school outbreaks [5,15,16]. However, some important factors, such as the timing of non-pharmaceutical measures, the total number of persons at risk, and the school’s district, were not adjusted for. As a result, the effectiveness of vaccines and non-pharmaceutical measures in school-based outbreaks remains unclear. To address this knowledge gap, we investigated the impact of influenza vaccination and non-pharmaceutical measures on febrile outbreaks caused by influenza A/B viruses in Beijing schools. We used febrile illness outbreak surveillance reports from 1 August 2018 to 31 July 2019 as our primary data source. We report the results of our investigation.

## 2. Materials and Methods

### 2.1. Study Design and Information Collection

The scope of our study was febrile illness outbreaks caused by influenza A/B virus in primary and secondary schools in Beijing reported between 1 August 2018 and 31 July 2019. Febrile illness outbreaks are reported and processed through the febrile illness outbreak surveillance system, which was developed and implemented by the Beijing Center for Disease Prevention and Control. The system covers all primary and secondary schools and most kindergartens, colleges, and hospitals in Beijing. Definitions used in Beijing’s system are as follows: (1) a febrile illness, caused by any etiology, is an individual with a measured or self-reported axillary temperature ≥37.5 °C; (2) influenza-like illness (ILI) is an individual presenting with a measured or self-reported axillary temperature ≥38 °C plus cough or sore throat; (3) a mild febrile outbreak is 5 to 9 febrile illnesses in a single classroom or department within a 2-day period; and (4) a serious febrile outbreak is 10 or more febrile illnesses in a single classroom or department within a 2-day period or 10 or more epidemiologically linked ILIs in a school, kindergarten, or hospital. The Beijing government conducted a school-based influenza vaccination program from 15 October to 30 November 2018, just prior to influenza season. Both trivalent inactivated influenza vaccines (IIV3) and quadrivalent inactivated influenza vaccines (IIV4) were provided to primary and secondary school students at no charge to their families.

Febrile illness outbreaks were reported to local district Centers for Diseases Prevention and Control (CDCs). We conducted field investigations and collected oral pharyngeal swabs with local CDC staff within 24 h of an outbreak report. We collected epidemiological information using a standardized questionnaire and continued to monitor outbreaks until they ended—defined as when no epidemiologically linked febrile illnesses were reported for 7 consecutive days. School information included the type of school (primary, secondary, or nine-year school), the number of students in the school, influenza activity in the district when the outbreak was reported, influenza vaccine coverage in the school, and the starting date for influenza vaccination in the school. Epidemiological information included the number of students in classes, their genders, illnesses by gender, influenza vaccination coverage levels of classes, dates of influenza vaccination of classes, dates of non-pharmaceutical interventions, dates of onset for the first case in the febrile illness outbreak, and ending dates of febrile illness outbreaks. Non-pharmaceutical interventions included isolation of individuals with febrile illness, encouragement to wear a mask, enhanced disinfection and ventilation of classrooms, and use of targeted class closures if necessary. Since antibodies against influenza are typically produced 14 days after vaccination, we considered influenza vaccination coverage of classes to be zero if the class vaccination date was less than 14 days prior to the illness onset date of first case [3,17]. During our investigations, we collected at least five oral pharyngeal swabs for mild outbreaks and at least 10 oral pharyngeal swabs for severe outbreaks. Priority for swab collection was given to symptomatic students.

We conducted a survey of all primary and secondary schools that did not have febrile illness outbreaks in the Beijing area between 1 August 2018 and 31 July 2019. We considered these schools as a control group for comparison with schools that had febrile illness outbreaks.

### 2.2. Influenza Identification

Oral pharyngeal swabs were tested for influenza virus by real-time reverse transcription polymerase chain reaction assay (real-time RT-PCR) in local CDC laboratories. These laboratories are managed by the Beijing CDC and use the testing protocol of the WHO Collaborating Center for Reference and Research on Influenza at the Chinese National Influenza Center [18]. If more than half of the specimens were positive for influenza nucleic acid, we considered the outbreak to be caused by influenza virus. We used RT-PCR to verify the influenza virus subtype or lineage.

### 2.3. Data Management and Statistical Analysis

Two investigators independently extracted data using a self-developed data extraction form. We calculated means and standard deviations to describe normally distributed continuous variables, medians and inter-quartile ranges to summarize non-normally distributed continuous variables, and percentages for categorical variables. Normally distributed data were analyzed with analysis of variance; the Wilcoxon test was use for non-normally distributed data. To account for hierarchical social structures, in which schools, as lower level, are nested in districts, as higher level, we developed a two-level multiple logistic regression model to identify factors associated with serious influenza outbreaks and a multi-level random effect zero-inflated negative binomial regression (RE-ZINB) model to identify correlations between potential explanatory variables and influenza outbreaks [19]. We used different vaccination coverage levels (from 10% to 90%) as cut-off points to iteratively calculate relative risks (RR) of outbreaks using the two-level RE-ZINB model. We explored correlations between vaccination coverage levels and outbreaks using the multi-stage regression model. All *p* values were based on two-sided statistical tests; statistical significance was defined as a *p*-value < 0.05. Data were entered in an Excel spreadsheet, and all analyses were completed with SAS University Edition (SAS Institute, Inc., Cary, NC, USA), R software Version 3.6.1 (R Foundation for Statistical Computing, Vienna, Austria), or Joinpoint Regression Program Version 4.8.0.1 (US National Cancer Institute, Rockville, MD, USA).

## 3. Results

### 3.1. Febrile Illness Outbreak Characteristics

From 1 August 2018 to 31 July 2019, a total of 748 febrile illness outbreaks were identified, involving 8176 individuals (febrile illnesses) in Beijing. Among the 748 febrile outbreaks, 407 were in primary schools (54.42%; 4491 febrile illnesses), 213 in kindergartens (28.48%; 2191 febrile illnesses), 106 in secondary schools (14.17%; 1237 febrile illnesses), and 22 in other settings (2.94%; 257 febrile illnesses). The median number of febrile illnesses per outbreak was 10, with an inter-quartile range from 8 to 13.

We obtained 5336 oral pharyngeal swabs in 731 outbreaks, among which 3070 (57.53%) were positive for influenza virus; 645 outbreaks (86.23%) were caused by influenza A/B viruses, including A(H1N1)pdm09 (150 outbreaks; 20.05%), A(H3N2) (167 outbreaks; 22.33%), B(Victoria) (295 outbreaks; 39.44%), B(Yamagata) (four outbreaks; 0.53%), or more than one influenza virus subtypes/lineages (29 outbreaks, 3.88%); 86 outbreaks were negative for influenza and 17 outbreaks were unable to be identified etiologically due to sampling failures (Figure 1).

### 3.2. Serious Outbreaks vs. Mild Outbreaks

Among the 748 febrile illness outbreaks, 462 were caused by influenza A/B virus in primary and secondary schools. Among these 462 outbreaks, 108 were serious outbreaks (54.42%; 1935 febrile illnesses) and 354 were mild outbreaks (54.42%; 3289 febrile illnesses). In univariate analysis, ten factors were compared between serious and mild outbreaks. As shown in Table 1, larger class size and more days between identification of the first case to initiation of an intervention were significantly associated with risk of serious febrile outbreaks (*p* < 0.05). Vaccination coverage levels of classes and schools were significantly lower in serious febrile illness outbreaks compared with mild outbreaks (*p* < 0.05).

The two-level logistic regression model identified correlations between potential explanatory factors and the size of outbreaks. In the empty model, the between-group variance of ${\hat{\sigma }}_{\mu 0}^{2}$ was statistically significant (χ^2^ = 15.15, *p* < 0.001) with an intra-class correlation coefficient (ICC) of 0.19, indicating a moderately large between-district heterogeneity. As shown in Table 2, after other factors were adjusted for in the two-level logistic regression model, large class size (odds ratio (OR) = 2.38; 95% CI: 1.28~4.26) and days from first case onset to intervention (OR = 1.17; 95% CI: 1.02~1.34) remained as risk factors and were significantly associated with serious outbreaks. Influenza virus subtypes or lineages had no association with serious outbreaks (*p* > 0.05). After other factors were adjusted for, there was no statistically significant association between high vaccination coverage (>50%) and serious outbreaks (OR = 0.87; 95% CI: 0.4~1.89).

### 3.3. Vaccination for School Influenza Outbreaks

From August 2018 to July 2019, 462 influenza outbreaks were reported in 283 (14.72%) primary and secondary schools; there were 1728 (85.28%) schools without outbreaks in the Beijing area. As shown in Table 3, the distribution of school type was significantly different in the two groups (*p* < 0.01). The number of classes and the number of students were significantly lower in schools without outbreaks (*p* < 0.01). School-level vaccination coverage was significantly higher in schools without outbreaks (*p* < 0.01); there was no significant difference by school area (rural or urban) (*p* = 0.67).

To model count data with extra zeros in the hierarchically structured data, we used a two-level RE-ZINB model to identify possible correlations between potential explanatory factors and influenza outbreaks. In the empty model, the between-group variance of ${\hat{\sigma }}_{\mu 0}^{2}$ was statistically significant (χ^2^ = 48.03, *p* < 0.001) with an intra-class correlation coefficient (ICC) of 0.11, indicating a moderately large between-district heterogeneity. Five potential explanatory factors were included in the model: high vaccination coverage of schools (>50%), more classes in schools (>18), more students in schools (>535), type of school, and location of school (urban or rural). After other factors were adjusted for in the two-level RE-ZINB model, high vaccination coverage (RR = 0.50; 95% CI: 0.34–0.75), as a protective factor, was significantly associated with febrile illness outbreak, implying that high vaccination coverage (>50%) was associated with having half as many influenza outbreaks. After other factors were adjusted for, schools with more students were more likely to have outbreaks (RR = 4.19; 95% CI: 1.98–8.87) and schools in urban areas were less likely to have outbreaks (RR = 0.28; 95% CI: 0.16–0.47).

To identify possible correlations between relative factors and outbreaks of influenza subtype/lineage, we tested three additional two-level RE-ZINB models to compare schools with outbreaks caused by an influenza virus subtype/lineage with other schools. As shown in Table 4, after other factors were adjusted for, high vaccination coverage (>50%) was associated with fewer outbreaks caused by three influenza virus subtypes/lineages, and the individual strain association was statistically significant for outbreaks caused by B (Victoria) influenza viruses (RR = 0.40; 95% CI: 0.24–0.67).

We used two-level RE-ZINB models to estimate the RR values iteratively using different vaccination coverage levels as cut-off points—from 10% to 90% coverage, with step sizes of 1%. Based on this series of cut-off points and the corresponding RR values, we used a multi-stage regression model (joinpoint) to explore correlations between vaccination coverage levels and outbreaks. As shown in Figure 2, RR values decreased with increasing vaccination coverage when vaccination coverage was greater than 30% (95% CI: 28~32%). When coverage was between 45% and 51%, RR values decreased the fastest, with an annual percent change (APC) of −6.00 (*p* = 0.01). When coverage was greater than 51% (RR = 0.42; 95% CI: 0.22–0.61), vaccination was associated with the largest protective effect against school influenza outbreaks.

## 4. Discussion

Using the Beijing CDC’s febrile illness outbreak surveillance system, we found 462 influenza outbreaks in primary or secondary schools during a one-year period, from August 2018 to July 2019. Our evaluation of these outbreaks showed that high influenza vaccine coverage in schools or classes and rapid identification and initiation of non-pharmaceutical interventions (NPIs) were associated with fewer and smaller school-based influenza outbreaks. Vaccine coverage over 50% was associated with a 50% reduction in the number of school-based influenza outbreaks. Our findings show the value of Beijing’s school-based febrile illness surveillance system for rapid initiation of NPIs to reduce the size of outbreaks and of Beijing’s policy of offering free influenza vaccination to school children for the prevention of influenza outbreaks.

In China, children attend school for approximately 8 h a day, providing ample close-contact opportunities for spreading influenza virus, leading to outbreaks. Since children are believed to be key introducers of influenza virus into communities [4,8,9], prevention of school outbreaks may help reduce community introduction and transmission of influenza.

High influenza vaccine coverage could prevent influenza outbreaks in primary and secondary schools, even if the vaccine is not well matched, which could reduce febrile outbreaks caused by influenza in primary and secondary schools by 50%. In the subgroup analysis for each subtype/lineage, the reduction was statistically significant only for B(Victoria) viruses. However, high vaccine coverage was linked to reduced outbreaks caused by A(H1N1)pdm09 or A(H3N2), which were not statistically significant. We speculated that there are two reasons leading to these results. The first one is that there were not enough samples to detect statistical differences. The second one might be vaccine mismatch. During the 2018–2019 influenza season, A(H1N1)pdm09, A(H3N2), and B(Victoria) co-circulated in the Beijing area [20]. Almost all A(H1N1)pdm09 viruses were antigenically matched with the vaccine virus, egg-propagated A/Michigan/45/2015 strain, and most A(H3N2) viruses were antigenically matched with the vaccine virus, egg-propagated A/Singapore/INFIMH-16-0019/2016 strain. Only a few of the B(Victoria) viruses were antigenically similar to the vaccine virus, egg-propagated B/Colorado/06/2017 strain, as B(Victoria) viruses had a three-amino-acid deletion in their Hemagglutinin (HA) antigens [21,22]. Because of a vaccine mismatch to B(Victoria), higher vaccine coverage was needed to avoid influenza outbreaks caused by B(Victoria), but a lower vaccine coverage could prevent outbreak caused by A(H3N2) or A(H1N1)pdm09 strains. A high vaccine coverage threshold may have concealed a relationship between vaccine coverage and outbreaks caused by the A(H1N1)pdm09 and A(H3N2) strains. On the other hand, as shown in Figure 2, the RR value of 22% vaccine coverage was lower than that of 30–40% vaccine coverage, which reflected that a lower vaccine coverage could prevent outbreaks by A(H3N2) or A(H1N1)pdm09.

In a multi-stage regression model, we found that when coverage was between 45% and 51%, the protective effect of the influenza vaccine increased rapidly, as each 1% increase in vaccination coverage was associated with a 6% reduction in outbreaks. A key result was that school vaccination coverage over 51% may be a threshold value to avoid school-based influenza outbreaks.

In the univariate analysis, we found that both early NPI intervention and high vaccine coverage could reduce the size of influenza outbreaks. However, after other factors were adjusted for in a two-level logistic regression model, only early intervention was significantly protective against serious outbreaks. We speculate that early identification of febrile illnesses and rapid initiation of NPI measures (ideally within three days) can interrupt virus transmission and reduce the size of an outbreak. Such an NPI effect would be in addition to vaccination and may conceal an independent impact of vaccination.

Our study had some limitations. First, the data used in this study were based on surveillance, which may be incomplete. Confounding factors, such as age, pattern of behavior, and reporting bias, could not be controlled for. Second, there were many febrile outbreaks in kindergartens, but some key variables, such as vaccination coverage, were not collected for kindergartens, and therefore, outbreaks in kindergartens were not analyzed. Third, our study used only one year of surveillance and coverage data. Fourth, the students’ antibodies against vaccine were not tested. According to a previous systematic review based on randomized controlled trials, seroconversion rates against different subtypes/lineages were different, which were from 57% to 72% [23]. That should be under consideration. A well-designed larger and longer-duration study and a serological sampling survey as a supplement are needed to yield more robust evidence.

## 5. Conclusions

In summary, we showed that high influenza vaccine coverage may prevent influenza outbreaks in primary and secondary schools, even when the influenza vaccine is not well matched to circulating strains. Early non-pharmaceutical measures may interrupt virus transmission and reduce the size of school-based outbreaks. We believe that our study may also imply that for COVID-19 prevention, a vaccine and non-pharmaceutical interventions could be used in combination to prevent and control COVID-19, once a vaccine against SARS-CoV-2 is available.

## Figures and Tables

**Figure 1 vaccines-08-00714-f001:**
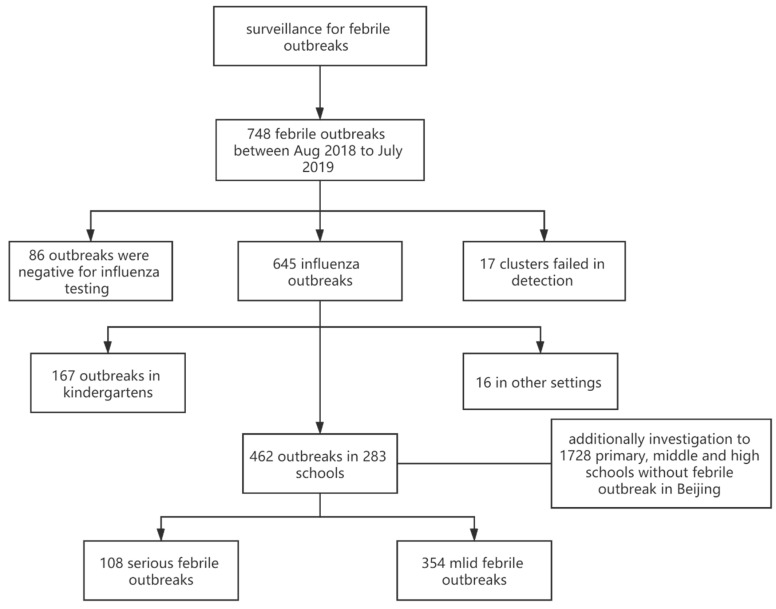
Flow chart and basic information of febrile outbreaks identified in Beijing from August 2018 to July 2019.

**Figure 2 vaccines-08-00714-f002:**
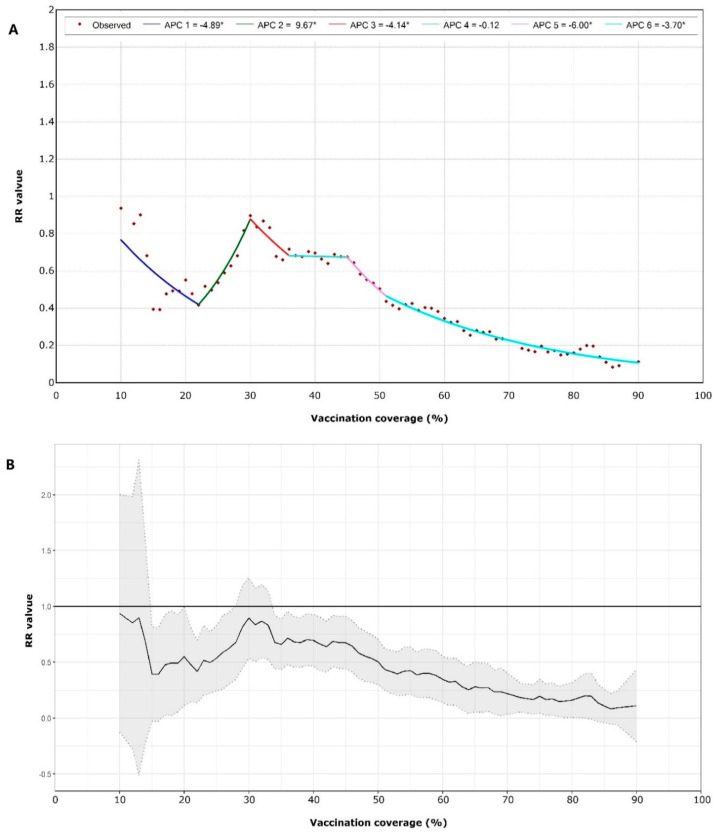
(**A**) Joinpoint model of relative risk (RR) values for different vaccination coverage. There are five joinpoints for the model. The symbol * is shown if the annual percent change (APC) is significantly different from zero at level 0.05. (**B**) RR values for vaccination coverage from 10% to 90% with 95% confidence interval.

**Table 1 vaccines-08-00714-t001:** Characteristic of febrile outbreaks in school in Beijing, from August 2018 to July 2019.

Characteristic	Mild Febrile Outbreaks (*n* = 354)	Serious Febrile Outbreaks (*n* = 108)	Total	Z/χ^2^	*p* Value
Type of school ^1^					
Primary school	282 (77.90%)	80 (22.10%)	362 (100.00%)	1.52	0.22
Middle/high school	72 (72.00%)	28 (28.00%)	100 (100.00%)		
Areas ^2^					
Urban	205 (78.54%)	56 (21.46%)	261 (100.00%)	1.24	0.27
Rural	149 (74.13%)	52 (25.87%)	201 (100.00%)		
Male/female in febrile illnesses ^2^	1.17 (0.67–1.67)	1.14 (0.79–1.42)	1.14 (0.67–1.67)	0.22	0.83
Male/female of class ^2^	1.07 (0.95–1.21)	1.10 (0.95–1.23)	1.07 (0.95–1.21)	0.55	0.58
Number of students in class ^2^	39.00 (34–43)	41.00 (37.0–45.5)	39.00 (35–43)	3.73	0.0002
Subtype/lineage ^1^					
B(Victoria)	173 (78.28%)	48 (21.72%)	221 (100.00%)	3.54	0.32
A(H1N1)pdm09	53 (82.81%)	11 (17.19%)	64 (100.00%)		
A(H3N2)	110 (72.37%)	42 (27.63%)	152 (100.00%)		
Other	18 (72.00%)	7 (28.00%)	25 (100.00%)		
Days from first case to intervention ^2^	3.00 (2.00–4.00)	4.00 (3.00–5.00)	3.00 (2.00–4.00)	4.84	<0.0001
Influenza activity when outbreak reported ^2,3^	0.40 (0.30–0.55)	0.44 (0.29–0.55)	0.40 (0.29–0.55)	0.44	0.66
Vaccination coverage of class ^2^	0.39 (0.24–0.52)	0.33 (0.21–0.47)	0.37 (0.23–0.51)	2.17	0.03
Vaccination coverage of school ^2^	0.38 (0.31–0.47)	0.36 (0.27–0.46)	0.38 (0.30–0.47)	1.98	0.047

^1^ Number of outbreaks and proportion were used to describe the variables, and the χ^2^ test was used to assess the difference between two groups. ^2^ Median and inter-quartile range were used to describe the variables and Wilcoxon’s test was used to assess the difference between two groups. ^3^ Influenza activity when outbreak reported: weekly positive proportion of influenza-like illness (ILI) for influenza in the district when outbreak was reported.

**Table 2 vaccines-08-00714-t002:** Two-level logistic regression model comparing serious outbreaks and mild outbreaks in Beijing, from August 2018 to July 2019.

Variables	*β*	*t*	*p*	OR (95%CI)
Intercept	−2.19	−3.61	0.003	0.11 (0.03–0.42)
Primary school	−0.27	−0.81	0.43	0.76 (0.37–1.54)
Urban	0.57	1.09	0.29	1.78 (0.57–5.44)
More male illnesses	−0.02	−0.08	0.94	0.98 (0.56–1.66)
More male students in class	0.26	0.93	0.37	1.29 (0.72–2.36)
Large class	0.87	3.08	0.008	2.38 (1.28–4.26)
High level influenza activity when outbreak reported	0.17	0.65	0.53	1.19 (0.68–2.14)
A(H1N1)pdm09 vs. A(H3N2)	−0.63	−1.44	0.17	0.53 (0.21–1.36)
B(Victoria) vs. A(H3N2)	−0.24	−0.72	0.48	0.79 (0.4–1.63)
Other vs. A(H3N2)	0.02	0.04	0.97	1.02 (0.33–3.21)
Days from first case to intervention	0.16	2.37	0.03	1.17 (1.02–1.34)
High vaccination coverage of class	−0.27	−0.85	0.41	0.77 (0.35–1.35)
High vaccination coverage of school	−0.10	−0.28	0.78	0.9 (0.4–1.89)

Notes: Large class: number of students in class was more than the median of the number of students in all classes with febrile outbreaks in Beijing, which was 39 students. High level influenza activity when outbreak reported: weekly positive proportion of ILI for influenza in the district when outbreak was reported was higher than 40%. High vaccination coverage: a vaccination coverage higher than 50%.

**Table 3 vaccines-08-00714-t003:** Characteristic of schools in Beijing, from August 2018 to July 2019.

Characteristic	School without Outbreak (*n* = 1728)	School with Outbreaks (*n* = 283)	Total	Z/χ^2^	*p*
Type of school ^1^					
Primary school	1010 (83.82%)	195 (16.18%)	1205 (100%)	39.94	<0.001
Nine-year school	53 (68.83%)	24 (31.17%)	77 (100%)		
Middle/high school	665 (91.22%)	64 (8.78%)	729 (100%)		
Number of classes in school ^1^	17 (10–25)	26 (18–37)	18 (11–28)	10.57	<0.001
Number of students in school ^1^	488 (235.5–844)	910 (574–1395)	535 (269–910)	12.01	<0.001
Vaccination coverage of school ^1^	0.49 (0.34–0.67)	0.42 (0.32–0.53)	0.48 (0.33–0.65)	−5.36	<0.001
Areas ^2^					
Urban	915 (86.24%)	146 (13.76%)	1061 (100%)	0.18	0.67
Rural	813 (85.58%)	137 (14.42%)	950 (100%)		

^1^ Median and inter-quartile range were used to describe the variables and Wilcoxon’s test was used to assess the difference between two groups. ^2^ Number of schools and proportion were used to describe the variables and the χ^2^ test was used to assess the difference between two groups.

**Table 4 vaccines-08-00714-t004:** Effect of vaccine on febrile outbreaks caused by different influenza virus subtype/lineage, based on a two-level random effect zero-inflated negative binomial (RE-ZINB) regression model.

Subtype/Lineage	Outbreaks	School with Outbreaks	Effect of High Vaccination Coverage (RR)	95%CI	*t*	*p*
A(H3N2)	152	114	0.78	0.20–3.10	−0.38	0.71
B(Victoria)	221	144	0.40	0.24–0.67	−3.8	0.002
A(H1N1)pdm09 ^1^	64	51	0.59	0.20–1.72	−1.06	0.31
Other ^2^	25	23	-	-	-	-
Total ^3^	462	283	0.50	0.34–0.75	−3.65	0.003

^1^ Because the two-level RE-ZINB model was not a convergence model, a two-level random effect zero-inflated Poisson regression (RE-ZIP) model was used to estimate parameters instead. ^2^ Because there were too few schools with other influenza subtypes/lineages, both the two-level RE-ZINB and RE-ZIP models were not convergence models. ^3^ There had been outbreaks caused by different subtypes/lineages in 49 schools.

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
