# Peer review of "Influenza Vaccination and Non-Pharmaceutical Measure Effectiveness for Preventing Influenza Outbreaks in Schools: A Surveillance-Based Evaluation in Beijing"

_vaccines, 2020, doi:10.3390/vaccines8040714_

Round 1
Reviewer 1 Report
This is a well written, interesting, and useful contribution, which I think is entirely suitable for publication in Vaccines. Although I am intimately familiar with the topic of this paper, I found Table 4 is difficult to follow; I suspect a reader less familiar with the topic might have even greater difficulties. What does Influenza mean?
Other minor points:
Line185: This sentence does not flow. Some words missing between “of” and “is”.
Line217: Figure 1 should be Figure 2.
Author Response
Dear reviewer,
We are truly thankful to you for the questions and suggestions that are so helpful making our paper "Influenza vaccination and non-pharmaceutical measures in preventing outbreaks in school based on surveillance systems in Beijing." (vaccines-965435) more suitable for publication. The changes to the manuscript have been reviewed and approved by all the authors. Our responses to your comments are as follow.
1, Although I am intimately familiar with the topic of this paper, I found Table 4 is difficult to follow; I suspect a reader less familiar with the topic might have even greater difficulties. What does Influenza mean?
Answer: As your suggestion, Table 4 have been revised to make it easier to understand, and the information regarding this subject has been added to the Results section of the revised manuscript (Line 259-270). “Influenza” in Table 4 means outbreaks caused by any subtype/lineage of influenza, and the confusing expression has been revised in Table 4.
2, Line185: This sentence does not flow. Some words missing between “of” and “is”.
Answer: As your suggestion, we have added mathematical symbol “” between “of” and “is” in revised manuscript (Line 196 and Line 248).
3, Line217: Figure 1 should be Figure 2.
Answer: As your suggestion, Figure 1 has been revised to Figure 2 in the revised manuscript (Line 269). Actually, the original manuscript in Word version was correctly displayed as “Figure 2”,however, in PDF version was displayed as “Figure 1”.
Thank you again for your advice. I hope I can learn more from you.
With best regards,
Yours sincerely
Reviewer 2 Report
This paper examines the causes of influenza epidemics in primary and secondary schools and the epidemic control effects of vaccination using statistical methods.
The method is very well scrutinized, the results are very precise, and the considerations for it are very well documented.
It is acceptable in this version.
Author Response
Dear reviewer,
We are truly thankful to you for the comments that are so helpful making our paper " Influenza vaccination and non-pharmaceutical measures in preventing outbreaks in school based on surveillance systems in Beijing." (vaccines-965435) more suitable for publication. The changes to the manuscript have been reviewed and approved by all the authors.
Thank you again for your advice. I hope I can learn more from you.
With best regards,
Yours sincerely
Reviewer 3 Report
Dear Editor-in-Chief,
I have now read the manuscript Vaccines-965435 entiteled: “Influenza vaccination and non-pharmaceutical measures in preventing outbreaks in school based on surveillance systems in Beijing.” by Sun Ying et al. The manuscript cover an important area of preventive health activity involving school children and respiratory infection with influenza. Unfortunately, the manuscript in its current condition is quite confusing why the tables and the interpretation of significant differences of the table-presented data could be explained more clearly.
Comments:
Q1. Language need thorough corrections.
Questions:
Q2 How did the students respond serologically to the vaccine? How many of them reached antibody titers that confirm protective levels?
Q3.If antibody reactivity was not analyzed, then perhaps the study resulted in changed behaviour rather than efficient vaccination explaining the reduced incidence.?
A vaccine interference more favorable behaviour, with more social distancing?
Q4. Placebo- or for example pertussis-vaccination as an option to influenza vaccine, with same favourable result?
Results
Q5. Line 211. The authors refer to a Figure 2, but this figure seem to be missing in the manuscript?
Discussion
Q6. Line 233-240. The authors discuss the lack of significant reduction in influenza outbreaks. This is not that surprising since the level of vaccination introduced seldom reached above 50%. Thus there may be a trend towards reduced influenza outbreak when vaccine coverage was >70% (as illustrated in Figure 1), but it is still not significant. Conclusion could be that vaccine interference during an outbreak have no significant preventive effect?
Q7. Figure 1. Is it so that the figure show that a 22% vaccine coverage is more efficient in RR-value reduction than 30-40% vaccination coverage? Please explain.
Tables:
Q8. Table 3. There are several calculation errors in this table? These errors need to be corrected, and recalculations may result in different results than these presented in the current Table 3. (i.e Numbers of students in school, and Numbers of classes in school are summarized erroneously?)
Author Response
Dear reviewer,
We are truly thankful to you for the questions and suggestions that are so helpful making our paper "Influenza vaccination and non-pharmaceutical measures in preventing outbreaks in school based on surveillance systems in Beijing." (vaccines-965435) more suitable for publication. The changes to the manuscript have been reviewed and approved by all the authors. Our responses to your comments are as follow.
Q1. Language need thorough corrections.
Answer: As your request, the language of this manuscript has been revised.
Q2. How did the students respond serologically to the vaccine? How many of them reached antibody titers that confirm protective levels?
Answer: We didn’t test antibody against vaccine. According to previous systematic review based on randomized controlled trials, seroconversion rates are 71.63%, 68.45%, 57.28%, and 63.12% against A(H1N1)pdm09, A(H3N2), B(Victoria), and B(Yamagata), respectively, at day 21 post-vaccination. Your valuable suggestion is a limitation of our study, we will conduct a serological sampling survey as a supplement in the future. The information about this subject has been added to the Discussion section of the revised manuscript (Line 348-351).
Q3. If antibody reactivity was not analyzed, then perhaps the study resulted in changed behaviour rather than efficient vaccination explaining the reduced incidence.?A vaccine interference more favorable behaviour, with more social distancing?
Answer: Thanks for your valuable suggestion. Our study was based on an underlying assumption, which was children stayed in school for approximately 8 h per day that was long enough for virus spreading. However, non-pharmacological measures for COVID-19 showed pattern of behavior had a greater effect against virus spreading. The information about this subject has been added to the Discussion section of the revised manuscript (Line 341-343).
Q4. Placebo- or for example pertussis-vaccination as an option to influenza vaccine, with same favourable result?
Answer: Influenza viruses are easy to occur antigenic variation, especially among influenza A viruses. Influenza vaccine should be injected every year. It’s hard to find a vaccine/placebo as influenza vaccine’s control to be vaccinated every year.
Q5. Line 211. The authors refer to a Figure 2, but this figure seem to be missing in the manuscript?
Answer: As your suggestion, Figure 1 has been revised to Figure 2 in the revised manuscript (Line 269). Actually, the original manuscript in Word version was correctly displayed as “Figure 2”,however, in PDF version was displayed as “Figure 1”.
Q6. Line 233-240. The authors discuss the lack of significant reduction in influenza outbreaks. This is not that surprising since the level of vaccination introduced seldom reached above 50%. Thus there may be a trend towards reduced influenza outbreak when vaccine coverage was >70% (as illustrated in Figure 1), but it is still not significant. Conclusion could be that vaccine interference during an outbreak have no significant preventive effect?
Answer: Maybe our expression misled you. High influenza vaccination coverage could prevent influenza outbreaks in primary and secondary schools, even vaccine was not well matched, which could reduce 50% febrile outbreaks caused by influenza in primary and primary and secondary schools. We changed expression in the Discussion section, renewed Table 4, and added a new figure to Figure 2 of the revised manuscript to make the article more understandable (Line 290-309, Line 248-254, Line 268-272). As showed in Table 3, the median of vaccination coverage in primary and secondary schools was 48%, the value in schools without outbreak was 49%. As showed in Figure 2, vaccine could reduce influenza outbreaks with statistically significant when vaccination coverage was >70%.
Q7. Figure 1. Is it so that the figure show that a 22% vaccine coverage is more efficient in RR-value reduction than 30-40% vaccination coverage? Please explain.
Answer: As showed in Figure 2.B of revised manuscript, the RR value of 22% vaccination coverage was lower than 30-40% vaccination coverage. Because vaccine well matched A(H3N2) or A(H1N1)pdm09 and mismatched B(Victoria) in 2018-2019 influenza season. In this situation, higher vaccination coverage was needed to avoid influenza outbreaks caused by B(Victoria) and lower vaccination coverage could prevent outbreak caused by A(H3N2) or A(H1N1)pdm09. The cut-off point of higher vaccination coverage may conceal the relationship between vaccine and outbreaks caused by A(H1N1)pdm09 and A(H3N2). The information about this subject has been added to the Discussion section of the revised manuscript (Line 290-309).
Tables:
Q8. Table 3. There are several calculation errors in this table? These errors need to be corrected, and recalculations may result in different results than these presented in the current Table 3. (i.e Numbers of students in school, and Numbers of classes in school are summarized erroneously?)
Answer: We recalculated twice and found no error in Table 3.
Thank you again for your advice. I hope I can learn more from you.
With best regards,
Round 2
Reviewer 3 Report
Dear Editor-in-Chief,
I have now read the revised manuscript Vaccines-965435 entiteled: “Influenza vaccination and non-pharmaceutical measures in preventing outbreaks in school based on surveillance systems in Beijing.” by Sun Ying et al. The manuscript cover an important area of preventive health activity involving school children and respiratory infection with influenza. Unfortunately, the manuscript in its current condition is less but still a bit confusing.
Comments:
Language has been significantly improven.
Questions
Q1 How did the students respond serologically to the vaccine? How many of them reached antibody titers that confirm protective levels?
The authors presented the answer that they have in the reply, but have not added this information into the manuscript. Why not?
Author Response
Dear reviewer,
We are truly thankful to you for the attentive review and questions that are so helpful making our paper "Influenza vaccination and non-pharmaceutical measure effectiveness for preventing influenza outbreaks in schools, a surveillance-based evaluation in Beijing." (vaccines-965435) more suitable for publication. The changes to the manuscript have been reviewed and approved by all the authors. Our responses to your comments are as follow.
Q1 How did the students respond serologically to the vaccine? How many of them reached antibody titers that confirm protective levels?
The authors presented the answer that they have in the reply, but have not added this information into the manuscript. Why not?
Answer: Due to our negligence, this information was not added into the revised manuscript. We have added it to Discussion section of the revised manuscript (Line 283).
Thank you again for your advice. I hope I can learn more from you.
With best regards,
Yours sincerely